# Lysosomal Acid Lipase as a Molecular Target of the Very Low Carbohydrate Ketogenic Diet in Morbidly Obese Patients: The Potential Effects on Liver Steatosis and Cardiovascular Risk Factors

**DOI:** 10.3390/jcm8050621

**Published:** 2019-05-07

**Authors:** Stefano Ministrini, Lucia Calzini, Elisa Nulli Migliola, Maria Anastasia Ricci, Anna Rita Roscini, Donatella Siepi, Giulia Tozzi, Giulia Daviddi, Eva-Edvige Martorelli, Maria Teresa Paganelli, Graziana Lupattelli

**Affiliations:** 1Internal Medicine, Department of Medicine, Università degli Studi di Perugia, 06129 Perugia, Italy; luciacalzini@hotmail.it (L.C.); elisanullimigliola@gmail.com (E.N.M.); m.anastasia.ricci@gmail.com (M.A.R.); arita.roscini@ospedale.perugia.it (A.R.R.); donatella.siepi@unipg.it (D.S.); giuliadaviddi@yahoo.it (G.D.); eva.martorelli@libero.it (E.-E.M.); graziana.lupattelli@unipg.it (G.L.); 2Hepatology, Gastroenterology and Nutrition Unit, IRCCS “Bambino Gesù” Children’s Hospital, 00165 Rome, Italy; giulia.tozzi@opbg.net; 3General Surgery, “Santa Maria della Misericordia” Hospital, 06129 Perugia, Italy; chir.gen.silvestrini@ospedale.perugia.it

**Keywords:** ketogenic diet, morbid obesity, lysosomal acid lipase, non-alcoholic fatty liver disease, lipid metabolism

## Abstract

A very low carbohydrate ketogenic diet (VLCKD) is an emerging technique to induce a significant, well-tolerated, and rapid loss of body weight in morbidly obese patients. The low activity of lysosomal acid lipase (LAL) could be involved in the pathogenesis of non-alcoholic fatty liver disease (NAFLD), which is a common feature in morbidly obese patients. Fifty-two obese patients suitable for a bariatric surgery intervention underwent a 25-day-long VLCKD. The biochemical markers of glucose and lipid metabolism, and flow-mediated dilation (FMD) of the brachial artery were measured before and after VLCKD. LAL activity was measured using the dried blood spot technique in 20 obese patients and in a control group of 20 healthy, normal-weight subjects. After VLCKD, we observed a significant reduction in body mass index, fasting glucose, insulinemia, and lipid profile parameters. No significant variation in FMD was observed. The number of patients with severe liver steatosis significantly decreased. LAL activity significantly increased, although the levels were not significantly different as compared to the control group. In conclusion, VLCKD induces the activity of LAL in morbidly obese subjects and reduces the secretion of all circulating lipoproteins. These effects could be attributed to the peculiar composition of the diet, which is particularly poor in carbohydrates and relatively rich in proteins.

## 1. Introduction

Fatty liver disease is a very common feature of obesity [1] and it significantly contributes to the surgical risk involved in bariatric interventions, since it increases the risk of intra-operatory mechanical liver damage [2].

A very low carbohydrate ketogenic diet (VLCKD) is an emerging technique to induce a significant, well-tolerated, and rapid loss of body weight in morbidly obese patients [3]. VLCKD induced weight loss is faster than the weight loss induced by a balanced low-caloric diet, while the subsequent weight gain is not faster than the weight gain following a low-caloric diet [4,5,6,7]. 

Previous studies demonstrated an amelioration of glucose metabolism and endothelial function, measured as flow-mediated dilation (FMD), in obese patients after VLCKD; however, the results are still conflicting about the effects on FMD [4,5,6,7,8,9,10].

VLCKD has been recently used to bring about immediate preoperative weight loss in patients suitable for bariatric surgery interventions. Indeed, it simplifies anesthesiologic and surgical maneuvers in these patients and it reduces the risk of pre- and post-operative complications. Moreover, the degree of operative risk reduction seems to be related to the amount of weight loss [11]. Since the amount of weight loss obtained with VLCKD is related to the improvement in liver volume and steatosis [12], it has been hypothesized that the efficacy of pre-operative VLCKD is, at least in part, attributable to its effect on liver steatosis [13].

Lysosomal acid lipase (LAL) plays a central role in lipid metabolism in humans. Congenital LAL deficiency, caused by mutations at the LAL locus on chromosome 10q23.2 (more than 40 loss-of-function mutations have been identified so far) is a rare autosomal recessive disease, known as Wolman’s syndrome and is characterized by the accumulation of cholesterol esters and triglycerides in all cells, primarily in the liver and spleen, though other organs are involved [14]. The late onset of this disease, known as cholesteryl-ester storage disease (CESD), is seen as having a wide clinical spectrum with recognition of the diagnosis ranging from childhood to adulthood. The clinical manifestations are commonly hepatomegaly/liver steatosis, elevation of liver enzyme, and dyslipidemia [15,16].

Previous studies have hypothesized that an acquired LAL deficiency could be involved in the pathogenesis of non-alcoholic fatty liver disease (NAFLD). Baratta et al. [17] investigated LAL activity in a cohort of adult patients with NAFLD compared to healthy subjects. They found a significant reduction in LAL activity in NAFLD patients and the enzyme activity was significantly worse in the subgroup of patients with biopsy-proven non-alcoholic steatohepatitis (NASH). More recently, Tovoli et al. [18] demonstrated that this deficiency is specific to NAFLD by comparing NAFLD patients with HCV-infected patients, particularly in the pre-cirrhotic stage of disease. 

Since to our knowledge there are no data related to the modulation of LAL activity through the weight loss induced by VLCKD, we hypothesize that the effects of VLCKD on liver steatosis could be mediated by an induction of this enzyme.

Therefore, the aim of our study was to evaluate the effect of a 25-day-long VLCKD on liver steatosis and LAL activity in a group of morbidly obese patients suitable for bariatric surgery. We also evaluated if a relationship exists between the effects of VLCKD on LAL activity and other anthropometric, cardiovascular, and metabolic modifications induced by VLCKD in morbidly obese subjects.

## 2. Experimental Section

### 2.1. Study Population

Fifty-two obese patients suitable for a bariatric surgery intervention, referred to the Obesity Clinic of Università degli Studi di Perugia for a pre-operative evaluation, were enrolled in the study. Recruitment started on 1 October 2016 and ended on 1 December 2017.

Subjects included in the study were aged between 18 and 65 years with a BMI ≥ 40 kg/m^2^ or a BMI ≥ 35 kg/m^2^ and obesity related comorbidities (type 2 diabetes mellitus, systemic arterial hypertension, obstructive sleep apnea syndrome, etc.).

Patients with type 1 diabetes mellitus, severe chronic kidney disease (K-DOQI 3/5 or above), severe heart and liver failure (NYHA IV and Child–Pugh B or above, respectively), or severe psychiatric disorders were excluded from the study. 

All measurements were collected at baseline up to 14 days before the beginning of the diet, and once again, up to seven days after the end of the diet. Venous and capillary blood samples were all collected in the morning after a 12-hour fast.

A group of 20 healthy, normal-weight subjects (mean age 43 ± 13, mean BMI 22.8 ± 2.6 kg/m^2^) was used as control group. A further group of 20 obese subjects with grade 1 obesity and no related comorbidity (10 males and 10 females, mean age 48 ± 9 years, mean BMI 30.4 kg/m^2^) was used as a sham control group. These patients were given only nutritional and lifestyle advices at the moment of enrolment. LAL activity values were measured at the moment of enrolment and after 30 days.

All procedures were in accordance with the ethical standards of the institutional and/or national research committee and with the 1964 Helsinki declaration and its later amendments or comparable ethical standards. This study was approved by the local ethical board of Università degli Studi di Perugia (2014-020) and all study subjects signed an informed consent form to voluntarily participate in this study. The study was registered at ClinicalTrials.gov with the following registration number: NCT03564002.

### 2.2. Diet Composition

Subjects underwent a 25-day planned VLCKD with caloric restriction (<800 kcal/day). Carbohydrate intake was <50 g/day (corresponding to <200 kcal/day) with a protein intake of 1.4 g/kg of ideal weight (calculated with the Lorentz formula [19]):

Ideal weight (male) = (height in cm – 100) − (height in cm – 150)/4;

Ideal weight (female) = (height in cm – 100) − (height in cm – 150)/2.

Assuming an average ideal weight of 65 kg for both women and men, the protein intake was >90 g/day (corresponding to >350 kcal/day). The remaining caloric intake was composed of fat (<250 kcal/day, corresponding to <30 g/day).

The protein intake was achieved in part with a dietary supplement composed of milk whey protein (Nepicomplex, 39 g/day corresponding to 156 kcal/day) diluted in water or skimmed milk. The remaining part was achieved with fresh aliments in appropriate quantities, depending on ideal body weight. Details of the diet are shown in Figure 1.

The dietary supplements mentioned above have the following compositions:

Solus multinutrient (1 tablet)—calcium carbonate (160 mg), calcium phosphate (105 mg), iron gluconate (14 mg), zinc gluconate (12.5 mg), magnesium oxide (60 mg), vitamin C (60 mg), choline bitartrate (20 mg), lecithin powder (6 mg), rutin, p-aminobenzoic acid (25 mg), vitamin B2 (1.6 mg), alfalfa leaves, Siberian ginseng, biotin (0.15 mg), inositol (15 mg), vitamin E (10 mg), alga kelp, primrose oil, copper gluconate (1 mg), vitamin A (800 mcg), vitamin B5 (18 mg), selenium methionine (30 mcg), betaine chloride, citrus flavonoids, hesperidin, manganese sulphate (1 mg), vitamin B6 (2 mg), folic acid (200 mcg), vitamin D (5 mcg), vitamin B1 (1.4 mg), watercress powder, parsley powder, rice bran, vitamin B12 (1 mcg), potassium chloride (200 mg), amino acid chelate chromium, potassium iodide, sodium molybdenum.

Xalifom (1 tablet)—calcium citrate (400 mg), magnesium citrate (150 mg), vegetal cellulose, sodium citrate (150 mg), potassium citrate (500 mg), iron gluconate (20 mg), magnesium stearate (150 mg), copper gluconate (1 mg), manganese gluconate (0.5 mg), silica micronized, folic acid (100 mcg).

All patients were also invited to compile a daily diary to make note of any physical symptoms (e.g., headache, constipation, nausea, vomiting, weakness, dizziness, muscle cramps, palpitations) and hunger. Hunger was self-reported by means of the three-factor eating questionnaire [20]. Water intake was fixed to no less than 2 liters/day.

Self-testing of urinary ketones was administrated once daily. Urine ketone concentrations were measured using over-the-counter reagent strips (Accu-Chek Ketur Test, Roche Diagnostics GmbH, Mannheim, Germany), which determine the presence of ketones upon reaction with nitroprussiate salt. Urine ketones were assessed using a semi-quantitative scale. The compliance of patients to VLCKD was verified by the presence of ketones in the urine.

### 2.3. Serum Measurement of Biochemical Markers

At baseline and after 25 days of VLCKD, the following parameters were evaluated: blood count (flow-cytometry, DxH 800 AU, CoreLab, Beckman Coulter Italia, Milan, Italy), glutamic oxaloacetic transaminase (GOT), glutamic pyruvic transaminase (GPT), gamma glutamyl transpeptidase (γGT) (chemiluminescence, DxC 700 AU, CoreLab, Beckman Coulter Italia, Milan, Italy), total and high-density lipoprotein (HDL)-cholesterol, triglycerides (enzymatic colorimetric assay, DxC700, CoreLab, Beckman Coulter Italia, Milan, Italy), fasting glucose, and insulin (immune-enzymatic assay, UniCel DxI 800, Beckman Coulter Italia, Milan, Italy).

The homeostasis model of insulin resistance (HOMA-IR) was used as a measure of insulin resistance. 

### 2.4. Lysosomal Acid Lipase Activity Assay

In a subgroup of 20 patients (6 males and 14 females, mean age 53 ± 9, mean BMI 44 ± 8 kg/m^2^), LAL activity was measured before and after the diet. 

LAL activity was measured with the dried blood spot (DBS) technique using the inhibitors Lalistat 2 as reported by Hamilton et al. [21].

Ethylene-diamine-tetra acetic acid (EDTA) blood, obtained by venepuncture, was spotted onto filter paper (Whatman grade 903 Schleicher & Schuell) and allowed to dry overnight at room temperature. 

Samples were stored with desiccant at −20 °C and analyzed within 1 month of storage. Activities were measured after being uninhibited and inhibited with Lalistat 2, and LAL activity was determined by subtracting the activity in the inhibited reaction from the uninhibited reaction (total lipase) and expressed as nmol/spot/h of 4 MU (methylumbelliferone). DBS tests were performed at “Bambino Gesù Hospital” in Rome (Italy). 

### 2.5. Bioimpedentiometry

Fat mass, expressed as a percentage of body weight, was measured through bioimpedentiometry (50 kHz, amplitude 50 mA, Body Composition Analyzer TBF-410GS; Tanita, Tokyo, Japan) with electrodes applied on the plantar surface of both feet. 

### 2.6. Ultrasonography

Ultrasonographic measurements were performed by trained operators (known here as M.A.R. and L.C.), who were blinded to laboratory values. L.C. performed the liver steatosis assessment and visceral fat area measurement, while M.A.R. performed FMD measurements. Repeated measurements were performed by the same operator each time. The collection of the images and their interpretation were performed by the same operator. Previous studies reported an intra-observer agreement ranging 51–68% for the ultrasonographic assessment of liver steatosis [22], 94% for visceral fat area [23], and 84–99% for FMD [24].

Abdominal ultrasonography with a 3.5 MHz convex probe (MyLab 50; Esaote, Genoa, Italy) was used to estimate the visceral fat area (VFA), expressed in cm^2^, and the severity of liver steatosis. The visceral fat area was estimated according to the Hirooka formula [25].

Liver steatosis was assessed with a semi-quantitative method, as previously described by Joseph et al. [26], and was based on the presence of three qualitative criteria: parenchymal hyper-echogenicity, compared to the kidney cortical echogenicity; posterior beam attenuation with standard settings; and blurred visualization of intrahepatic vessels and diaphragm (grade 0 = no steatosis, grade 1 = mild steatosis, grade 2 = moderate steatosis, grade 3 = severe steatosis).

Arterial ultrasonography with a 7.5 MHz (MyLab 50; Esaote, Genoa, Italy) linear probe was used to measure the FMD at the brachial artery of the non-dominant arm, as previously described by Thijssen et al. [27]. 

### 2.7. Statistical Analysis

Analyses were performed using SPSS software for Windows (version 17.0; SPSS, Inc., Chicago, IL, USA), with significance set at 2-sided *p* < 0.05.

Values are expressed as mean ± standard deviation for continuous variables and as numbers (%) for discrete variables. Normal distribution of variables was tested with the Kolomogorov–Smirnov non-parametric test. The Student t test and the one-way ANOVA were used to test the signifance of the differences between groups. The Student t-paired test and Wilcoxon test were used to test the significance of the variation before and after VLCKD for the parametric and non-parametric variables, respectively. Variations were calculated by subtracting post- and pre-diet parameters. The difference in distributions was tested with the χ^2^ test. Correlation coefficients were calculated with Spearman correlation rank tests. A quadratic regression model was employed.

## 3. Results

The characteristics of 52 obese patients before and after VLCKD are listed in Table 1. Eighteen subjects (35%) were male and 34 (65%) were female with a mean age of 49 ± 12.5 years. Nine patients (17.6%) had an established diagnosis of diabetes and 37 patients (73%) had an established diagnosis of systemic arterial hypertension before enrolment; of these, three patients (6%) were on oral antidiabetic treatment and nine patients (17%) were on anti-hypertensive treatment. No patient was taking lipid-lowering drugs.

Characteristics of the control group are summarized in Table 2.

All patients had a positive test for urinary ketones up to 72 h after the beginning of the diet. No patient abandoned the diet because of excessive hunger or unacceptable physical symptoms.

After VLCKD, body weight and BMI were significantly reduced (*p* < 0.001 for both), yet significantly higher than the control group (*p* < 0.001 for both). Similarly, waist circumference, fat mass, and VFA were significantly reduced as well (*p* < 0.001 for each). Among the glucose metabolism parameters, we observed a significant reduction in fasting glucose levels (*p* = 0.007), insulinemia (*p* < 0.001), and HOMA-IR (*p* < 0.001). After VLCKD, average levels of fasting glucose reverted to the normal range (60–100 mg/dL [28]) yet were significantly higher than the control group (*p* = 0.021). 

Lipid profile parameters were also significantly reduced: a 23% reduction in triglycerides (*p* < 0.001), a 17% reduction in total cholesterol (*p* < 0.001), a 10% reduction in HDL-cholesterol (*p* = 0.002), and a 22% reduction in LDL-cholesterol (*p* < 0.001) were observed, although the levels of total cholesterol and HDL-cholesterol were in the reference range (HDL-cholesterol: 40–50 mg/dL for males, 50–59 mg/dL for females; total cholesterol: <200 mg/dL for both genders [29]). Furthermore, total cholesterol and LDL-cholesterol were not significantly different compared to the control group (*p* = 0.183 and *p* = 0.223, respectively).

No significant variation in FMD was observed. 

We also observed a significant improvement in liver steatosis. The number of patients with grade 3 steatosis decreased from 22 to 12, with a parallel increase in the number of patients with grade 1 steatosis from 10 to 20 (Table 1). Levels of γGT were significantly reduced after VLCKD, while the levels of GOT and GPT increased, yet inside the reference range (7–55 IU/L for GPT, 8–48 IU/L for GOT [30]).

The characteristics of subjects whose levels of lysosomal acid lipase activity were measured are summarized in Table 3.

As displayed in Figure 2, LAL activity in obese patients significantly increased after VLCKD (*p* = 0.012), although its levels were not significantly different compared to the control group.

Subjects in the sham control group experienced a slight but significant decrease in average BMI values (29.9 ± 2.1 kg/m^2^ vs. 30.4 ± 2.2 kg/m^2^, *p* = 0.008) with no significant change in LAL activity (1.08 ± 0.35 vs 1.06 ± 0.34, *p* = 0.204). No significant difference in LAL activity values at baseline was detected among the three groups (*p* = 0.358). 

After performing a univariate correlation analysis, a significant positive correlation was observed between LAL activity and body weight (r = 0.499, *p* = 0.025), BMI (r = 0.490, *p* = 0.028), waist circumference (r = 0.487, *p* = 0.029), and fat mass (r = 0.527, *p* = 0.020). A significant negative correlation between LAL activity and body weight was also observed in the control group (r = −0.0563, *p* = 0.010).

As displayed in Figure 3, taking the patients and control group together, we observed a significant quadratic relationship between LAL activity and body weight (beta1 = −0.058, *p* < 0.001, *R*^2^ = 0.338), as well as LAL activity and BMI (beta1 = −0.123, beta2 = 0.002, *p* = 0.039, *R*^2^ = 0.161).

No significant correlation was found between the improvement of LAL activity and the changes in other parameters, including anthropometric measures, bioimpedentiometry, degree of steatosis, glucose homeostasis, and lipid profile.

## 4. Discussion

In the present study, a 25-day VLCKD in morbidly obese patients resulted in a significant reduction in body weight, BMI, waist circumference, and fat mass. These changes were accompanied by significant decreases in various parameters related to glucose and lipid metabolism and in the degree of liver steatosis. LAL activity also significantly increased. A significant correlation between measures of adiposity (body weight, BMI, waist circumference, and fat mass) and LAL activity was observed.

Our results show for the first time that VLCKD induces the activity of LAL in morbidly obese subjects, although its levels in this population were not significantly different when compared to a group of healthy, normal-weight subjects. To our knowledge, no previous study has ever demonstrated a modulation of the activity of LAL in humans, so no data are available on the epigenetic modulation of LAL activity in vivo in humans.

We also observed an improvement in the degree of liver steatosis, as well as in the features of glucose homeostasis. The improvement in the degree of liver steatosis was associated with a significant reduction in γGT, which is commonly considered a marker of cellular damage in the context of NAFLD [31]. However, no significant correlation was detected between the improvement in liver steatosis and the reduction in γGT. Conversely, GOT and GPT underwent a significant increase; however, since their levels remained inside the reference range, we cannot consider this increase as being clinically relevant. Notwithstanding, this observation might be important in reference to the safety issues for treatments lasting more than three to four weeks.

Safety of ketogenic diets has been largely discussed in the literature. Most of the existing studies have been performed in children with refractory epilepsy, who are usually treated with normocaloric ketogenic diets for long periods; this notwithstanding short term adverse events are usually mild in both children and adults [32]. Since some adverse effects have been attributed to the chronic acidosis, supplementation with alkalinizing minerals is routinely used [33]. Since the VLCKD in obese patients is usually employed to achieve a significant weight loss immediately before bariatric surgery, the diet doesn’t usually last more than three to four weeks. No significant adverse effect has been reported in the literature for short courses of VLCKD in obese patients [12] and our results are consistent with these previously reported data.

The effects of VLCKD on liver steatosis and glucose metabolism are well documented in the literature [34] and our results are consistent with the previous studies.

The effects we observed on the lipid profile were unexpected; indeed, previous studies on morbidly obese patients, undergoing weight loss through surgical and non-surgical methods, reported a reduction in serum triglycerides, an increase in HDL-cholesterol and no significant variation in total cholesterol and LDL-cholesterol [35,36]. In our group of patients undergoing VLCKD, we observed a general reduction in all circulating lipoproteins, with triglycerides and LDL-cholesterol undergoing the largest relative reduction. 

We did not observe a significant variation in FMD mean values. Although most previous studies in obese subjects reported a significant improvement in FMD after VLCKD [8,9], a meta-analysis conducted by Schwingshackl et al. [10] reported a significant impairment. The lack of statistical significance in our results prevents us from supporting one of these two hypotheses; this might be attributable to the short duration of the observation (four to five weeks).

The improvement of LAL activity obtained through VLCKD was accompanied by weight loss, liver steatosis improvement, and an amelioration of glucose and lipid metabolism. However, none of these parameters significantly correlated with the increase in LAL activity, so we cannot identify a potential factor responsible for this improvement. 

LAL activity improvement could be caused by the weight loss induced by VLCKD. However, no significant difference was observed between the obese patients and healthy controls, and as can be seen in Figure 3, the values of LAL activity after VLCKD overwhelm the values of the healthy, normal-weight controls, although their BMI is still significantly higher than the controls. Furthermore, a positive correlation between LAL activity and the markers of adiposity (body weight, BMI, waist circumference, and fat mass) was observed in obese subjects. Since a negative correlation between LAL activity and body weight was observed in the control group, we hypothesized a U-shaped relationship between LAL activity and adiposity; this hypothesis was confirmed by the regression analysis.

This phenomenon could be hypothetically explained by a substrate-dependent enzymatic induction mechanism: in other words, triglycerides accumulating in the fatty liver of obese patients could induce the transcription of LAL genes. However, no correlation was observed between the degree of liver steatosis and LAL activity. This lack of statistical significance could be attributed to the limitations of the semi-quantitative ultrasonographic evaluation of liver steatosis, which is not a direct measure of liver fat content. Therefore, further studies involving more accurate diagnostic techniques (e.g., quantitative ultrasonography, MRI spectrometry, or CT scans) are necessary to confirm this hypothesis. Similarly, further in vitro studies are needed to confirm the transcriptional effects of triglycerides or their metabolites on LAL genes.

Furthermore, we should take into account the possibility of an independent effect of VLCKD on LAL activity. Indeed, its peculiar composition, particularly poor in carbohydrates and relatively rich in proteins, could modulate the liver metabolism of lipids acting on different molecular pathways, including LAL. This effect could explain the efficacy of VLCKD in reducing liver steatosis [37,38]. Indeed, previous studies reported an association between reduced LAL activity and NAFLD [39], and consistently, we observed a significant improvement in the degree of liver steatosis after VLCKD, although no significant correlation between the improvement of steatosis degree and variation of LAL activity was observed. Moreover, this lack of significance could be explained by the limitations of the ultrasonographic measurement of liver steatosis, as discussed above. 

This hypothesis is also consistent with the observation as regards the unexpected effects of VLCKD on the serum lipid profile, characterized by a significant reduction in all circulating lipoproteins. 

From a general point of view, VLCKD might be able to shift the liver lipid metabolism from storage to lipolysis in obese patients. This provisional conclusion is supported by the observation of the transcriptional effects of some essential amino acids in a murine model of the ketogenic diet. Indeed, previous studies have demonstrated that a diet enriched with five essential amino acids (Leu, Ile, Val, Lys, and Thr) prevents liver steatosis in mice fed with a high fat content diet [40] and it is associated with an increased expression of intracellular markers of mitochondrial function [41] and autophagy [42].

The main limitation of our study is the low number of patients, which could have prevented us from detecting some significant correlations. Furthermore, the presence of a second morbidly obese control group undergoing a low caloric balanced diet would have been useful to discriminate the effects of weight loss from the specific effects of VLCKD. In addition, the image readers were not blinded to the subjects’ group identity. Finally, as stated above, we did not measure the liver fat content.

## 5. Conclusions

In conclusion, we hypothesize that VLCKD is able to enhance LAL activity in morbidly obese subjects; this effect could, in turn, contribute to the healing of NAFLD. Furthermore, our results suggest that VLCKD has a wider effect on the liver lipid metabolism of morbidly obese subjects, shifting it from storage to lipolysis. The molecular mechanisms underlying these effects need to be explored and clarified in further studies.

## Figures and Tables

**Figure 1 jcm-08-00621-f001:**
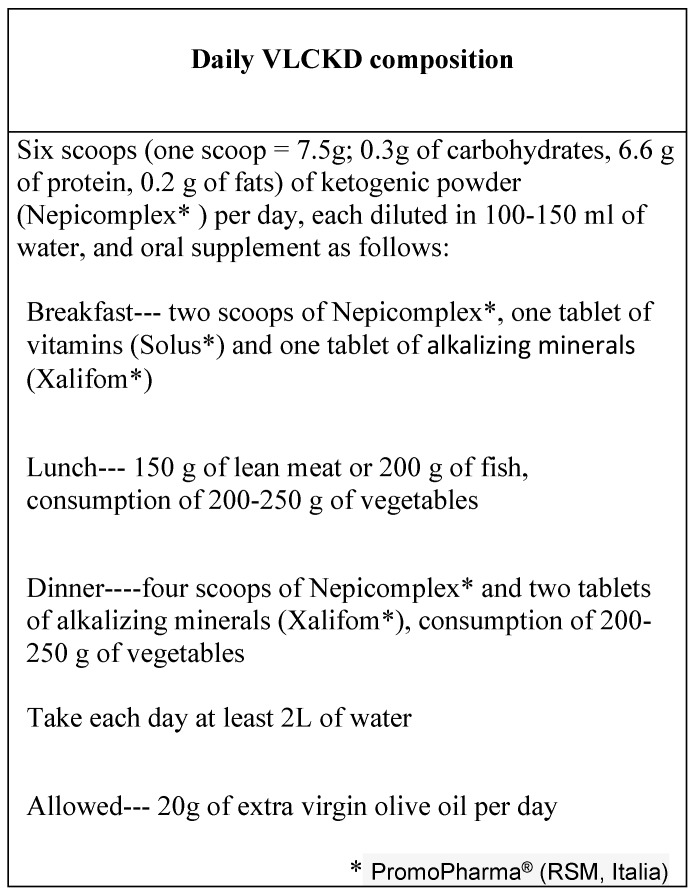
Daily very low carbohydrate ketogenic diet (VLCKD) composition.

**Figure 2 jcm-08-00621-f002:**
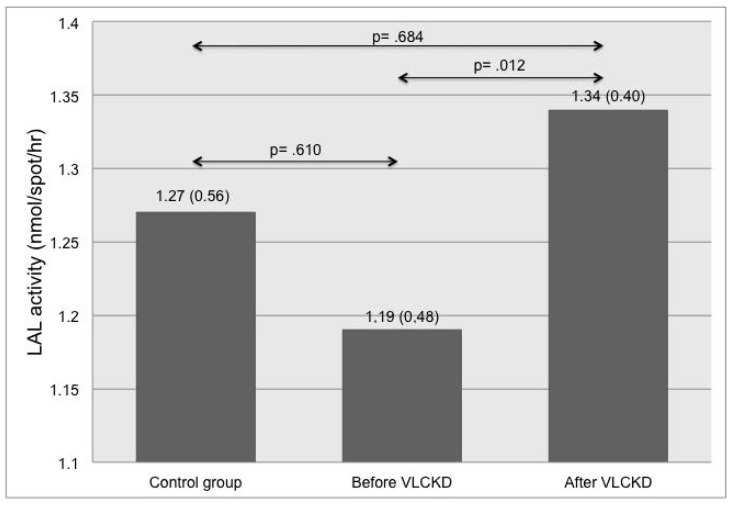
Differences in lysosomal acid lipase (LAL) activity among healthy subjects and obese patients before and after VLCKD.

**Figure 3 jcm-08-00621-f003:**
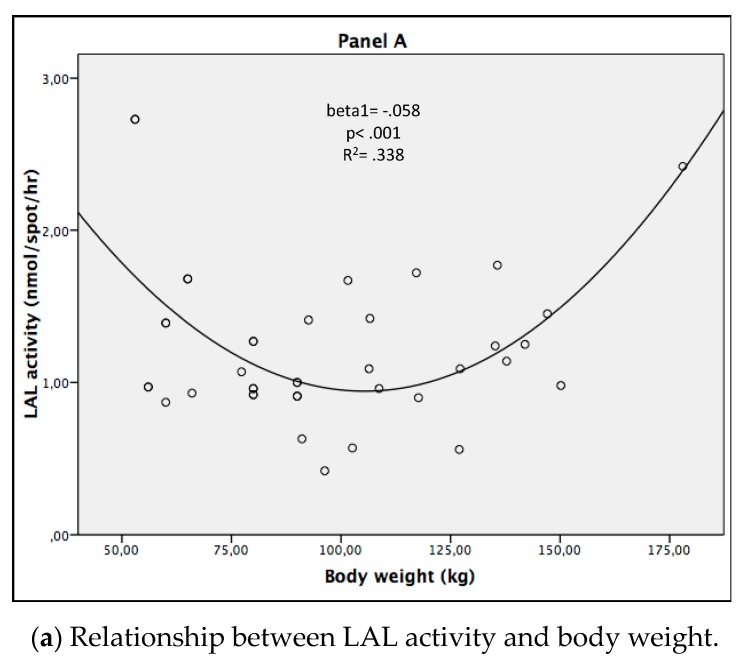
Quadratic relationship of LAL activity with body weight and BMI.

**Table 1 jcm-08-00621-t001:** Anthropometric, clinical, biochemical, and ultrasonographic characteristics at baseline and after VLCKD. Continuous values are expressed as mean ± standard deviation; distributions are expressed as numbers (%).

Characteristics of Patients	Baseline (52 pts)	After VLCKD (52 pts)	*p*-Value
**Clinical Data**			
Weight, kg	122 ± 24	115 ± 23	<0.001
Body mass index, kg/m^2^	44.7 ± 8.3	42 ± 8	<0.001
Systolic blood pressure, mmHg	130 ± 16	122 ± 15	<0.001
Diastolic blood pressure, mmHg	80 ± 16	74 ± 10	0.004
Waist circumference, cm	130 ± 19	124 ± 17	<0.001
Hip circumference, cm	135 ± 15	127 ± 23	<0.001
**Bioimpedance Analysis**			
Fat mass, %	47 ± 7	45 ± 6	<0.001
Fat mass, Kg	57 ± 14	52 ± 13	<0.001
**Biochemical Parameters**			
Total cholesterol, mg/dL	191 ± 30	159 ± 25	<0.001
LDL cholesterol, mg/dL	110 ± 21	88 ± 20	<0.001
HDL cholesterol, mg/dL	50 ± 15 M: 42 ± 11 F: 54 ± 16	45 ± 12 M: 40 ± 12 F: 48 ± 11	0.002 0.321 0.006
Triglycerides, mg/dL	155 ± 95	121 ± 63	<0.001
GOT, IU/L	26.4 ± 13	29.1 ± 13	0.037
GPT, IU/L	32.6 ± 20	39.3 ± 27	0.016
γGT, IU/L	38.5 ± 31	25.9 ± 14	<0.001
Fasting blood glucose, mg/dL	105 ± 29	95 ± 18	0.007
Insulin, mU/mL	23.4 ± 22.6	14.3 ± 10.5	<0.001
HOMA index	6.3 ± 7.1	3.6 ± 3.3	<0.001
**Ultrasound Assessment**			
Visceral fat area, cm^2^	264 ± 74	235 ± 79	<0.001
Flow-mediated dilation, %	11.98 ± 5.7	12 ± 5	0.920
**Liver Steatosis**			
No steatosis	3 (5.9)	4 (7.8)	<0.001
Grade 1	10 (19.6)	20 (39.2)	
Grade 2	16 (31.4)	15 (29.4)	
Grade 3	23 (43.1)	12 (23.5)	

LDL—low-density lipoprotein; HDL—high-density lipoprotein; GOT—glutamic oxaloacetic transaminase; GPT—glutamic pyruvic transaminase; γGT—gamma glutamyl transpeptidase; HOMA—homeostasis model assessment.

**Table 2 jcm-08-00621-t002:** Anthropometric and biochemical characteristics of the control group (20 subjects). *p*-values refer to the comparison with the patient group at baseline.

Characteristics of the Control Group	Mean ± SD	*p*-Value
Age, years	46.6 ± 13.5	0.467
Body weight, kg	72 ± 14	<0.001
Body mass index, kg/m^2^	23.5 ± 2.7	<0.001
Fasting blood glucose, mg/dL	84 ± 13	0.003
Total cholesterol, mg/dL	180 ± 34	0.183
Triglycerides, mg/dL	107 ± 45	0.037
LDL cholesterol, mg/dL	102 ± 27	0.223
HDL cholesterol, mg/dL	56 ± 13 M: 45 ± 10 F: 66 ± 4	0.147 0.626 <0.001
GOT, IU/L	22 ± 6	0.046
GPT, IU/L	21 ± 7	<0.001
γGT, IU/L	27 ± 14	0.142

LDL—low-density lipoprotein; HDL—high-density lipoprotein; GOT—glutamic oxaloacetic transaminase; GPT—glutamic pyruvic transaminase; γGT—gamma glutamyl transpeptidase.

**Table 3 jcm-08-00621-t003:** Characteristics of subjects whose levels of lysosomal acid lipase activity were measured before and after VLCKD. Continuous values are expressed as mean ± standard deviation; distributions are expressed as numbers (%).

	Baseline (20 pts)	After VLCKD (20 pts)	*p*-Value
Weight, kg	120 ± 25	113 ± 23	0.001
Body mass index, kg/m^2^	44 ± 8	41 ± 8	<0.001
Visceral fat area, cm^2^	281 ± 76	251 ± 80	0.009
Flow-mediated dilation, %	13 ± 6	12 ± 5	0.711
LDL-cholesterol, mg/dL	112 ± 26	98 ± 22	0.002
Triglycerides, mg/dL	184 ± 120	136 ± 74	0.003
Insulin, IU/L	22.8 ± 22.4	14.4 ± 7.8	0.007
HOMA index	5.6 ± 6	3.2 ± 2	0.009
γGT, IU/L	41.2 ± 30	28.8 ± 15	0.008
**Liver Steatosis**			
No steatosis	0 (0%)	2 (10%)	0.046
Grade 1	3 (15%)	7 (35%)	
Grade 2	4 (20%)	3 (15%)	
Grade 3	13 (65%)	8 (40%)	

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
