# Peer review of "Lysosomal Acid Lipase as a Molecular Target of the Very Low Carbohydrate Ketogenic Diet in Morbidly Obese Patients: The Potential Effects on Liver Steatosis and Cardiovascular Risk Factors"

_jcm, 2019, doi:10.3390/jcm8050621_

Round 1
Reviewer 1 Report
The authors have done an excellent job in making the revisions.
I have only 1 remaining issue (with 2 parts) -- the regression model is still not adequately described in the Methods and Results sections.
On page 7, line 200 it is stated "A quadratic regression model was employed." Suggest stating "A non-linear quadratic regression model was employed [to model the relationships of BMI and body weight with LAL activity] based on [fit statistics compared with linear models or whatever the rationale was for using non-linear modelling].
2. On page 10, lines 260-262, for the two models (one for body weight and one for BMI) that are both quadratic, it is not clear why beta1 and beta2 are reported for one model and only beta1 is reported for the other. Please report beta2 for the other model.
Author Response
Response to Reviewer 1
The authors have done an excellent job in making the revisions.
I have only 1 remaining issue (with 2 parts) -- the regression model is still not adequately described in the Methods and Results sections.
Comment #1
On page 7, line 200 it is stated "A quadratic regression model was employed." Suggest stating "A non-linear quadratic regression model was employed [to model the relationships of BMI and body weight with LAL activity] based on [fit statistics compared with linear models or whatever the rationale was for using non-linear modelling].
Response #1
We accept your suggestion and we accordingly modified the text. The sentence "A quadratic regression model was employed" was substituted by the sentence "A non-linear quadratic regression model was employed to model the relationships of BMI and body weight with LAL activity based on the results of correlation analysis" (Page 7, Line 199).
Comment #2
On page 10, lines 260-262, for the two models (one for body weight and one for BMI) that are both quadratic, it is not clear why beta1 and beta2 are reported for one model and only beta1 is reported for the other. Please report beta2 for the other model.
Response #2
Only beta1 was reported in the panel A because beta2 = 0 in the regression model and we thought that including a null coefficient in the legenda could be redundant. However, we accept your suggestion and we added the required information in the Figure 2, Panel A.
Reviewer 2 Report
The Authors followed the suggestions and improved the manuscript, however, still English language should be checked.
Author Response
The draft has already been revised by an english native speaker provided by the editor.
Reviewer 3 Report
none
Author Response
Thanks for your approval!
This manuscript is a resubmission of an earlier submission. The following is a list of the peer review reports and author responses from that submission.
Round 1
Reviewer 1 Report
This manuscript “Very Low Carbohydrate Ketogenic Diet in Morbid Obese Patients: Effects on Liver Steatosis, Lysosomal Acid Lipase, Glucose Metabolism and Endothelial Function” describes a study of changes in multiple metabolic and anthropometric parameters in 52 morbidly obese patients treated with the diet prior to bariatric surgery. Lysosomal acid lipase activity was measured in a subset of 20 of these patients and compared to 20 controls. The study is well-conducted and the results are interesting. Below are some suggestions for improving the presentation.
Major
1. With so many measurements being made in several different cohorts (52 obese patients and then 20 control vs. 20 obese), the presentation has to be much more clear and deliberate. The hypothesis being tested needs to be laid out more clearly in the Abstract and Introduction, with presentation of the Results and the Discussion following the same building of arguments.
Suggested flow (current manuscript does some of this, but it needs to be very linear): NAFLD occurs at a high rate in morbidly obese individuals. Weight loss occurring with very low carbohydrate ketogenic diets have been shown to improve the metabolic profile associated with morbid obesity, with some studies showing improvement in NAFLD. Low levels of lysosomal acid lipase occurring in morbidly obese individuals have been suggested to be involved in the pathogenesis of NAFLD. [All these arguments in current Intro can be included.] We therefore hypothesized that VLCKD may stimulate LAL which would be associated with a decrease in NAFLD morbidly obese patients. We also looked at the effects of VLCKD on glucose metabolism and endothelial cell function. (Why?) Need to fold these into the hypothesis. Are there other known or potential or questionable effects of VLCKD?
In summary, there needs to be much more clarity about the primary associations of interest (which seems to be VLCDK -> increase LAL -> decrease NAFLD as this is novel; other effects of VLCDK seem not to be) and how all of the other measurements fit into the overall hypothesis, particularly re: glucose and lipid metabolism and endothelial function as measured by FMD.
2. Lines 141-145: Measurement of liver steatosis: Were these assessments made by one reader? If not, was degree of agreement measured? Were readers blind to the subject’s group identity (control vs. obese? Pre-diet vs. post-diet?).
3. More explicit detail about regression models, for example, what the beta’s stand for and why only beta1 for body weight and beta 1 and 2 for BMI? How did you decide on a quadratic model? Seems like it would be better to look at change in body weight and change in LAL. Lines 211-213 seem to imply that this was done for the other parameters—would state this as “change in LAL and change in other parameters” since “improvement” has not been defined.
4. Figure 1: Suggest replacing dynamite plots with plots showing all points and adding connecting lines between before and after measurements.
5. Correlations: Suggest looking at change in LAL and change in body weight.
6. All measures detailed in Methods should be reported in Results. This would include:
a. Lines 101-104: Results of daily diary – including if the weight loss and BMI results are based on self-reported weight, -- if this is the case, it should be highlighted higher up in the article. Answers to the other questions should also be reported in the Results, even if just to say there were no significant changes in physical symptoms and hunger during the diet period.
b. Lines 105-109: Results of the compliance to the diet as determined by presence of ketones in the urine should be reported.
Minor (copy editing and word choices)
1. Suggest title change so as to either 1) highlight the LAL/liver histology results or 2) make it more broad rather than listing 4 specific things that were measured.
2. Suggest simply stating “control group” instead of using “CG” – there are already a lot of abbreviations in this article.
3. Consistent use of Very Low Carbohydrate [vs. Carbohydrates] Ketogenic Diet (Title vs. Line 15, Line 34, etc.)
4. Line 37: Not sure what “weight retrieval” means – “weight regain?” The opposite being “weight loss maintenance?”
5. Suggest consistent use of “operative” vs. “operatory” (Lines 41, 43, 46, etc.)
6. Line 64: Not sure what “amelioration of glucose metabolism and endothelial function” means. “Alteration?” instead of “amelioration”. Would be good to be specific here, i.e., what changes in glucose metabolism have been reported? What changes in FMD have been reported? Increase? Decrease?
7. Line 97: “proteic” should be “protein”
8. Line 115: Insulin(e) is misspelled.
9. Lines 119, 121: Would replace the word “dosed” with “measured.”
10. Line 178: Replace “normality” with “normal”
11. LAL activity “dosed” … does this mean “measured”?
12. Lines 242, 255: Replace the word “unfortunately” with “however” – the results are what they are.
13. Re: Discussion, the opening paragraph should re-state the results, something like: “VLCKD resulted in significant decreases in weight and BMI, which were accompanied by significant decreases in many measures of glucose metabolism (….) and lipid metabolism (…) and NAFLD in obese patients after 25 days on the diet. LAL activity was also significantly increased; changes in LAL in response to the diet were significantly correlated with …. However, we did not observe a significant correlation between diet-associated changes in LAL and changes in NAFLD.
(I may have some of the above incorrect, but perhaps that will help you see where there needs to be more clarity.)
Author Response
Reviewer #1
This manuscript “Very Low Carbohydrate Ketogenic Diet in Morbid Obese Patients: Effects on Liver Steatosis, Lysosomal Acid Lipase, Glucose Metabolism and Endothelial Function” describes a study of changes in multiple metabolic and anthropometric parameters in 52 morbidly obese patients treated with the diet prior to bariatric surgery. Lysosomal acid lipase activity was measured in a subset of 20 of these patients and compared to 20 controls. The study is well-conducted and the results are interesting. Below are some suggestions for improving the presentation.
Major
1. With so many measurements being made in several different cohorts (52 obese patients and then 20 control vs. 20 obese), the presentation has to be much more clear and deliberate. The hypothesis being tested needs to be laid out more clearly in the Abstract and Introduction, with presentation of the Results and the Discussion following the same building of arguments.
Suggested flow (current manuscript does some of this, but it needs to be very linear): NAFLD occurs at a high rate in morbidly obese individuals. Weight loss occurring with very low carbohydrate ketogenic diets have been shown to improve the metabolic profile associated with morbid obesity, with some studies showing improvement in NAFLD. Low levels of lysosomal acid lipase occurring in morbidly obese individuals have been suggested to be involved in the pathogenesis of NAFLD. [All these arguments in current Intro can be included.] We therefore hypothesized that VLCKD may stimulate LAL which would be associated with a decrease in NAFLD morbidly obese patients. We also looked at the effects of VLCKD on glucose metabolism and endothelial cell function. (Why?) Need to fold these into the hypothesis. Are there other known or potential or questionable effects of VLCKD?
In summary, there needs to be much more clarity about the primary associations of interest (which seems to be VLCDK -> increase LAL -> decrease NAFLD as this is novel; other effects of VLCDK seem not to be) and how all of the other measurements fit into the overall hypothesis, particularly re: glucose and lipid metabolism and endothelial function as measured by FMD.
Response #1
We thank you for your valuable contribution. We reorganized the flow of Introduction as you suggested:
- NAFLD in morbid obese subjects (Lines 35-37)
- General properties of VLCKD (Lines 38-44)
- Benefits of VLCKD in bariatric surgery (Lines 45-51)
- LAL biology (Lines 52-60)
- Role of LAL in NAFLD (Lines 61-67)
Afterwards, we tried to state our hypothesis in a clearer manner. So we added the following sentence at Lines 68-70: “Since to our knowledge there are no data about the modulation of LAL activity through the weight loss induced by VLCKD, we hypothesize that the effects of VLCKD on liver steatosis could be mediated by an induction of this enzyme”.
Then we stated the aim of our study in a clearer manner, declaring why measurements other than liver steatosis and LAL activity were included in the study. So we added the following sentence at Lines 71-75: “So, the aim of our study was to evaluate the effect of a 25 days-long VLCKD on liver steatosis and LAL-activity in a group of morbid obese patients suitable for bariatric surgery. We evaluated also if a relationship exists between the effects of VLCKD on LAL activity and other anthropometric, cardiovascular and metabolic modifications induced by VLCKD in morbid obese subjects”.
2. Lines 141-145: Measurement of liver steatosis: Were these assessments made by one reader? If not, was degree of agreement measured? Were readers blind to the subject’s group identity (control vs. obese? Pre-diet vs. post-diet?).
Response #2
We agree that our former description of ultrasonographic mesurements was incomplete. We consequently modified the sentence as follows: “Ultrasonographic measurements were performed by trained operators (M.A.R. and L.C.) who were blinded to laboratory values. L.C. performed liver steatosis assessment and visceral fat area measurement, while M.A.R. performed FMD measurements. Repeated measurements were performed by the same operator each time. Collection of the images and their interpretation were performed by the same operator. Previous studies reported an intra-observer agreement ranging 51-68% for ultrasonographic assessment of liver steatosis [22], 94% for visceral fat area [23] and 84-99% for FMD [24].” (Lines 162-168).
As it can be easily understood by this description, readers were not blind to the subject’s group identity. This has been reported among limitations (Lines 329-334).
3. More explicit detail about regression models, for example, what the beta’s stand for and why only beta1 for body weight and beta 1 and 2 for BMI? How did you decide on a quadratic model? Seems like it would be better to look at change in body weight and change in LAL. Lines 211-213 seem to imply that this was done for the other parameters—would state this as “change in LAL and change in other parameters” since “improvement” has not been defined.
Response #3
A quadratic regression model implies that the relationship between two variables can be described by a quadratic function, generally defined as y= beta1*x2+ beta2*x + alpha, with beta1 and beta2 being the coefficients and alpha being the intercept. If beta2= 0, then we have still a quadratic function with only one coefficient (oppositely, if beta1= 0 we have a simple linear function). However, in our opinion, introducing these mathematical concepts goes beyond the aim of the article.
We decided for a quadratic model because, as we state in the discussion (Lines 295-299) “a positive correlation between LAL activity and markers of adiposity (body weight, BMI, waist circumference and fat mass) was observed in obese subjects. Since a negative correlation between LAL activity and body weight was observed in the control group, we hypothesized a U-shaped relationship between LAL activity and adiposity; this hypothesis was confirmed by the regression analysis”. In the former version the word “negative” at Line 244 was wrong spelled as “positive”. However, in the description of results it was correctly stated that a negative correlation was found between LAL activity and body weight in the control group (Lines 287-290).
In Lines 211-213 (now 251-253) we state that “No significant correlation was found between the improvement of LAL activity and the changes in other parameters, including anthropometric measures, bioimpedentiometry, steatosis degree, glucose homeostasis and lipid profile”. This includes also the change in body weight, as an anthropometric measure. As you suggested, we changed the word “improvement” with “changes” in the current version.
4. Figure 1: Suggest replacing dynamite plots with plots showing all points and adding connecting lines between before and after measurements.
Response #4
We kindly appreciate your suggestion. We agree that a scatter dots plot would be better in communicating the entity of LAL activity variation before and after VLCKD. However this would prevent us from describing the differences with the control group, so we prefer to keep Figure 1 as it is. We hope you’ll understand our reason and agree with our decision.
5. Correlations: Suggest looking at change in LAL and change in body weight.
Response #5
As declared in response #3, “No significant correlation was found between the improvement of LAL activity and the changes in other parameters, including anthropometric measures, bioimpedentiometry, steatosis degree, glucose homeostasis and lipid profile” (Lines 251-353). This includes also the change in body weight, as an anthropometric measure.
6. All measures detailed in Methods should be reported in Results. This would include:
a. Lines 101-104: Results of daily diary – including if the weight loss and BMI results are based on self-reported weight, -- if this is the case, it should be highlighted higher up in the article. Answers to the other questions should also be reported in the Results, even if just to say there were no significant changes in physical symptoms and hunger during the diet period.
b. Lines 105-109: Results of the compliance to the diet as determined by presence of ketones in the urine should be reported.
Response #6
A typing mistake has occurred. Weight was not reported in the diary and the measurements we report have been collected in office.
As you suggest, we added the following sentence at Lines 209-211 “All patients had a positive test for urinary ketones up to 72 hours after the beginning of the diet. No patient dropped the diet because of excessive hunger or unacceptable physical symptoms”
Minor (copy editing and word choices)
1. Suggest title change so as to either 1) highlight the LAL/liver histology results or 2) make it more broad rather than listing 4 specific things that were measured.
Response #1
We accept your valuable suggestion, so we changed the title as follow: “Lysosomal Acid Lipase as a Molecular Target of Very Low Carbohydrate Ketogenic Diet in Morbid Obese Patients: Potential Effects on Liver Steatosis and Cardiovascular Risk Factors”
2. Suggest simply stating “control group” instead of using “CG” – there are already a lot of abbreviations in this article.
The suggested changes has been made.
3. Consistent use of Very Low Carbohydrate [vs. Carbohydrates] Ketogenic Diet (Title vs. Line 15, Line 34, etc.)
The suggested changes have been made.
4. Line 37: Not sure what “weight retrieval” means – “weight regain?” The opposite being “weight loss maintenance?”
As you suggested “Weight retrieval” has been modified as “weight regain” (Line 41)
5. Suggest consistent use of “operative” vs. “operatory” (Lines 41, 43, 46, etc.)
The suggested changes have been made (Lines 48, 50 and 79)
6. Line 64: Not sure what “amelioration of glucose metabolism and endothelial function” means. “Alteration?” instead of “amelioration”. Would be good to be specific here, i.e., what changes in glucose metabolism have been reported? What changes in FMD have been reported? Increase? Decrease?
Amelioration of glucose metabolism parameters stands for a reduction in parameters such as fasting glucose, insulinemia, HOMA index, HbA1c etc… Since different studies have used different outcomes, we chose this general definition.
Similarly, an amelioration of FMD means an increase of this parameter.
7. Line 97: “proteic” should be “protein”
The suggested change has been made
8. Line 115: Insulin(e) is misspelled.
The suggested change has been made
9. Lines 119, 121: Would replace the word “dosed” with “measured.”
The suggested changes have been made
10. Line 178: Replace “normality” with “normal”
The suggested change has been made
11. LAL activity “dosed” … does this mean “measured”?
See response to Comment #9
12. Lines 242, 255: Replace the word “unfortunately” with “however” – the results are what they are.
The suggested changes have been made
13. Re: Discussion, the opening paragraph should re-state the results, something like: “VLCKD resulted in significant decreases in weight and BMI, which were accompanied by significant decreases in many measures of glucose metabolism (….) and lipid metabolism (…) and NAFLD in obese patients after 25 days on the diet. LAL activity was also significantly increased; changes in LAL in response to the diet were significantly correlated with …. However, we did not observe a significant correlation between diet-associated changes in LAL and changes in NAFLD.
(I may have some of the above incorrect, but perhaps that will help you see where there needs to be more clarity.)
Response #13
We agree with your suggestion and we consequently added the following sentence at Lines 255-260: “In the present study, a 25-days VLCKD in morbid obese patients resulted in significant reduction of body weight, BMI, waist circumference and fat mass. These changes were accompanied by significant decreases in various parameters of glucose and lipid metabolism and in liver steatosis degree. LAL activity was also significantly increased. A significant correlation between measures of adiposity (body weight, BMI, waist circumference and fat mass) and LAL activity was observed”
Reviewer 2 Report
The clinical usefulness of the ketogenic diet in the treatment of obesity is the subject of debate. Not only the weight-loss effect but also the health effect of the diet is still the subject of discussion. In the present study the effect of ketogenic diet on weight loss, biochemical parameters related to lipid and glucose metabolism as well as lysosomal acid lipase activity and liver steatosis was assessed.
In the present study the ketogenic diet was presented as a very effective tool to significantly decrease body weight in morbidly obese patients. As morbidly obese patients are characterized by an inability to significant and effective loss of their body mass, a critical discussion of the study results and a critical discussion of the results of previous studies are needed. Biological relevance of the obtained statistically significant changes in assessed clinical and biochemical parameters is also expected.
More detailed description of patient characteristics is needed. It is mentioned that some study participants were diagnosed as diabetics. It can be also expected that in studied patients disturbances in lipid metabolism can also occur. However, the serum mean concentrations of lipids, glucose or insulin as well as blood pressure are in normal range. It indicates that at least some patients were under pharmacological treatment. Full and detailed patient’s characteristics should be presented.
The composition of the diet should be also more carefully described. What kind of vitamins were supplemented? What kind of minerals (tablet of alkalizing minerals ?) were supplemented? Daily intake of vitamins and minerals should be included. Did the Authors monitor daily intakes of meat, fish, oil and vegetables ?
It is not clear why the lysosomal acid lipase activity was not measured in all study participants but only in less than 40% of studied patients.
Results presented at figure 2 clearly indicate that in patients with similar body mass or similar BMI significant differences in LAL activity were observed. Therefore, data presented on fig. 2 should be discussed not only statistically analyzed.
The text should be corrected by a native speaker.
Author Response
Reviewer #2
The clinical usefulness of the ketogenic diet in the treatment of obesity is the subject of debate. Not only the weight-loss effect but also the health effect of the diet is still the subject of discussion. In the present study the effect of ketogenic diet on weight loss, biochemical parameters related to lipid and glucose metabolism as well as lysosomal acid lipase activity and liver steatosis was assessed.
Comment #1
In the present study the ketogenic diet was presented as a very effective tool to significantly decrease body weight in morbidly obese patients. As morbidly obese patients are characterized by an inability to significant and effective loss of their body mass, a critical discussion of the study results and a critical discussion of the results of previous studies are needed. Biological relevance of the obtained statistically significant changes in assessed clinical and biochemical parameters is also expected.
Response #1
The possibility to achieve a significant body weight reduction in morbid obese subjects, through behavioural, pharmacological and surgical interventions, has been well established in many large studies. See for instance the results of the Swedish Obesity Study, Ricci MA et al. Angiology 2016, Ricci MA et al. Angiology 2018 or Srivastava et al. Nat Rev Endocrinol 2018.
In particular, efficacy of VLCKD has been reported by several authors (i.e. Moreno et al. Endocrine 2014, Bueno et al. Br J Nutr, Leonetti et al. Obes Surg 2015, Yancy et al. Ann Intern Med 2004).
Yet, as you correctly state, despite an initial success of the treatments, weight regain is still a challenging problem in the treatment of obesity. Although this being a crucial issue, in our opinion discussing this problem goes far beyond the aim of our study.
Comment #2
More detailed description of patient characteristics is needed. It is mentioned that some study participants were diagnosed as diabetics. It can be also expected that in studied patients disturbances in lipid metabolism can also occur. However, the serum mean concentrations of lipids, glucose or insulin as well as blood pressure are in normal range. It indicates that at least some patients were under pharmacological treatment. Full and detailed patient’s characteristics should be presented.
Response #2
We agree with your comment and consequently we added a more detailed description of patients’ comorbidities in Results (Lines 191-196): “Nine patients (17.6%) had an established diagnosis of diabetes and 37 patients (73%) had an established diagnosis of systemic arterial hypertension before the enrolment; of them, 3 patients (6%) were on oral antidiabetic treatment and 9 patients (17%) were on anti-hypertensive treatment. No patient was assuming lipid lowering drugs”. Not surprisingly, no patient was assuming lipid lowering drugs because pure hypercholesterolemia is atypical in morbid obese subjects, whose typical lipid profile is characterized by high levels of triglycerides, low HDL-cholesterol and normal LDL-cholesterol.
Comment #3
The composition of the diet should be also more carefully described. What kind of vitamins were supplemented? What kind of minerals (tablet of alkalizing minerals ?) were supplemented? Daily intake of vitamins and minerals should be included. Did the Authors monitor daily intakes of meat, fish, oil and vegetables ?
Response #3
We agree with your comment and consequently we added a more detailed description of dietary supplements (Lines 113-123). Patients were educated to assume the correct food with appropriate quantities but their compliance to the diet was evaluated only through the detection of urinary ketones.
Comment #4
It is not clear why the lysosomal acid lipase activity was not measured in all study participants but only in less than 40% of studied patients.
Response #4
Since lysosomal acid lipase measurement is not included in the usual pre-operative assessment of our patients, we required a separate informed consent for this measurement. We included in the study all patients undergoing VLCKD in the time of observation, but only a part of them gave this additional consent.
Comment #5
Results presented at figure 2 clearly indicate that in patients with similar body mass or similar BMI significant differences in LAL activity were observed. Therefore, data presented on fig. 2 should be discussed not only statistically analyzed.
Response #5
We thank you for your valuable comment. Revising Figure 2, we noted that indeed one control has particularly high levels of LAL activity and his position in the scatter dot is particularly skewed from the fit-line, while all the other point lay around the fit-line. So, we performed the regression analysis once again, excluding this case and the regression models were still significant (p= .001 for BMI and p< .001 for body weight)
Comment #6
The text should be corrected by a native speaker.
Response #6
The current version has been corrected by the language assistance service provided by the editor. You can find the tracked changes in the text.
Reviewer 3 Report
Low activity of lysosomal acid lipase (LAL) could be involved in the pathogenesis of non-alcoholic fatty liver 17 disease (NAFLD), which is a common feature in morbid obese patients.
The authors investigated 52 obese patients before and after a 25-days long VLCKD. LAL activity was measured in a subgroup of the obese patients and in a control group of 20 healthy, normal-weight subjects.
As could be expected, reductions were observed after 25 days of VLCKD in body weight, BMI, waist circumference, fat mass and several glucose metabolism parameters.
LAL activity increased in the subgroup of patients after VLCD. However, an obese control group was not included. It cannot be excluded that he changes in LAL activity are due to variation in time, the VLCKD or the weight loss. It is also not clear why no significant difference was observed between obese patients and healthy controls in LAL activity. Therefore, the relevance and reliability of the LAL measurements is not clear.
No significant variation in flow mediated vasodilation was observed. However, it is not clear why the authors measured this and report this in the current paper.
The authors observed significant improvement in liver steatosis. However, ultrasound measured of liver steatosis is less reliable. The authors should explain why they believe that ultrasound is reliable.
Author Response
Reviewer #3
Low activity of lysosomal acid lipase (LAL) could be involved in the pathogenesis of non-alcoholic fatty liver 17 disease (NAFLD), which is a common feature in morbid obese patients.
The authors investigated 52 obese patients before and after a 25-days long VLCKD. LAL activity was measured in a subgroup of the obese patients and in a control group of 20 healthy, normal-weight subjects.
As could be expected, reductions were observed after 25 days of VLCKD in body weight, BMI, waist circumference, fat mass and several glucose metabolism parameters.
Comment #1
LAL activity increased in the subgroup of patients after VLCD. However, an obese control group was not included. It cannot be excluded that he changes in LAL activity are due to variation in time, the VLCKD or the weight loss. It is also not clear why no significant difference was observed between obese patients and healthy controls in LAL activity. Therefore, the relevance and reliability of the LAL measurements is not clear.
Response #1
We thank you for your valuable contribution. As it can be seen in Figure 2, patients and controls tend to lie on the two symmetrical branches of a parable. So, no significant difference can be detected between the two groups using a direct head-to-head confrontation.
Comment #2
No significant variation in flow mediated vasodilation was observed. However, it is not clear why the authors measured this and report this in the current paper.
Response #2
Measurement of flow-mediated dilation is a part of our usual pre-operative assessment of morbid obese patients. Since the efficacy of VLCKD induced weight loss on endothelial function is still debated, we wanted to give our contribution to this topic and we wanted to check if a relationship with LAL activity existed. We agree with you that the aim of the study is not clearly described, so we tried to explain it better in the introduction (Lines 68-75): “So, the aim of our study was to evaluate the effect of a 25 days-long VLCKD on liver steatosis and LAL-activity in a group of morbid obese patients suitable for bariatric surgery. We evaluated also if a relationship exists between the effects of VLCKD on LAL activity and other anthropometric, cardiovascular and metabolic modifications induced by VLCKD in morbid obese subjects”
Comment #3
The authors observed significant improvement in liver steatosis. However, ultrasound measured of liver steatosis is less reliable. The authors should explain why they believe that ultrasound is reliable.
Response #3
We thank you for your comment. Semiquantitative B-mode ultrasonographic grading of liver steatosis has poor sensibility and poor specificity if compared with CT scan, MRI or elastosonography. However, compared with these imaging techniques, it has some valuable advantages: indeed, it is cheap, fast, well validated (while elastosonography is not) and it doesn’t expose the patients to ionizing radiations (as CT scan does). For all these reasons, we preferred this imaging techniques, but we highlighted its limitations in the text at Lines 303-308 “However, no correlation was observed between the degree of liver steatosis and the activity of LAL. This lack of statistical significance could be attributed to the limitations of the semiquantitative ultrasonographic evaluation of liver steatosis, which is not a direct measure of liver fat content. So, further studies involving more accurate diagnostic techniques (e.g. quantitative ultrasonography, MRI spectrometry or CT scan) are necessary to confirm this hypothesis”, Lines 316-318 “Also this lack of significance could be explained by the limitations of the ultrasonographic measurement of liver steatosis, as discussed above” and Lines 333-334 “Ultimately, as stated above, we did not measure the liver fat content”.
Reviewer 4 Report
The manuscript titled, “Very Low Carbohydrate Ketogenic Diet in Morbid Obese Patients: Effects on Liver Steatosis, Lysosomal Acid Lipase, Glucose Metabolism and Endothelial Function” authored by Ministrini et al. investigated the effect of a very low carbohydrate ketogenic diet (VLCKD) on lysosomal acid lipase (LAL) activity which it’s low level of activity has been associated with the pathogenesis of non-alcoholic fatty liver disease (NAFLD). Obese subjects were placed on VLCKD for 25 days. Body weight, adiposity, fasting blood glucose and lipids, insulin sensitivity, and LAL activity were assessed before and after dietary intervention. Body weight, fasting blood glucose and lipids, and LAL activity were compared to a control group (CG) of 20 healthy, normal weight subjects. Following VLCKD, a significant reduction in BMI and fasting blood glucose and lipids were observed as well as improved insulin sensitivity. The degree of liver steatosis only decreased in the subjects with severe steatosis. LAL activity significantly increased following VLCKD, however the level of activity was not significantly different either before or after dietary intervention compared to CG. The authors conclude that VLCKD induces LAL activity in morbidly obese subjects.
The attention to lysosomes and LAL activity associated with obesity, inflammation, and NAFLD is significantly on the rise. Therefore, the investigation of the level of activity of LAL following dietary intervention is important. However, the current state of the study presented here is rather preliminary. There is not enough data to draw any significant conclusions . . . yet. The data presented are strong for preliminary data sets, however, sufficient power and proper controls are needed. The two largest weakness are: 1) The authors conclude that findings are specific to VLCKD. This could very well be true however the experimental design is not convincing as the findings could be due to the calorie restriction (800 kcal/day) alone. Therefore, an obese subject group, also suitable for bariatric surgery, should be studied following a dietary intervention of calorie restriction in which the diet consist of a more balanced macronutrient composition. 2) As no significant difference was observed in LAL activity either before or after VLCKD compared to the CG, the impact of the dietary invention on LAL activity is completely lost. Increased power of analysis could rescue this major flaw.
Of note, liver biopsies, collected at time of bariatric surgery, for histological and biochemical analysis could significant improve the impact of the study.
Furthermore, another major point of consideration is the use of the term ketogenic diet. The authors description of the diet is accurate as a very low carbohydrate diet will in fact generate ketones and a state of ketosis, as will calorie restriction, at least transiently. However, the ketogenic diet, in a general sense, is largely understood to be a low carbohydrate, high fat diet in which 20%, or less, of the calories are obtained from carbohydrates and 60%, or more, of the calories are obtained from fat. Therefore, it is highly recommended that authors reconsider using the term ketogenic. Although the ketogenic diet is trending in popularity, so is a low carbohydrate diet and is therefore of significant importance. The state of ketosis is however key to the current study as it suggest catabolic vs anabolic state of lipid metabolism.
Specific Points of Consideration
1) Text: It is highly recommended that the authors seek out editing and writing services prior to submission of manuscript review. The current layout does not have an organized flow and is therefore challenging to follow especially for a reader outside this area of research. The introduction is insufficient. More detailed information should be provided for points of interest as opposed to list of reports of association. It currently reads as though there was no specific aim. Emphasis should be placed on the importance of LAL activity. Why should we care? What are clinical consequences of LAL activity expression? Is it merely a correlation or is it a potential pharmacological target?
2) Table 1, Daily macronutrient composition would be better represented in an actual table format, listing macronutrient (carb, protein, and fat) content (mass and percentages) per meal and then totaled at bottom. This would make is clearer to the reader of the carefully executed experimental design.
3) Line 112 How were these parameters assessed? It is simply stated that they were measured. Please provide reference at minimum. Also, would be better described if details (samples were collected following 12-h fast, etc) were mentioned here rather than previously in this section. Also, were parameters assessed when patients were still on diet or afterwards? Line 82 states “up to 7 days after end of diet.” This is important. If measurements were taken days after end of diet, why? Seems like this would confound results and any differences could be missed.
4) Line 248 The authors state that a positive correlation was observed between LAL and markers of adiposity (body weight, BMI, waist circumference, and fat mass). Figure 2 only represents correlations in body weight and BMI yet the authors place emphasis on adiposity. The fat mass data from this study is available. If the authors wish to place emphasis on adiposity, a convincing correlation should be presented for fat mass as well.
5) Limitations to lack of significant differences in other parameters:
a. Low sample number
b. Short duration of diet
c. The fact that assessments were made up to 1 week after termination of the diet. Investigators may decrease the degree of variability if parameters are assessed while all patients are still on the diet. This would also simplify interpretation.
d. As the authors suggested, ultrasonographic measurement of liver steatosis.
e. Evidence that subjects remained in a daily state of ketosis. A low carbohydrate diet requires strict adherence to sustain ketosis.
Author Response
Reviewer #4
The manuscript titled, “Very Low Carbohydrate Ketogenic Diet in Morbid Obese Patients: Effects on Liver Steatosis, Lysosomal Acid Lipase, Glucose Metabolism and Endothelial Function” authored by Ministrini et al. investigated the effect of a very low carbohydrate ketogenic diet (VLCKD) on lysosomal acid lipase (LAL) activity which it’s low level of activity has been associated with the pathogenesis of non-alcoholic fatty liver disease (NAFLD). Obese subjects were placed on VLCKD for 25 days. Body weight, adiposity, fasting blood glucose and lipids, insulin sensitivity, and LAL activity were assessed before and after dietary intervention. Body weight, fasting blood glucose and lipids, and LAL activity were compared to a control group (CG) of 20 healthy, normal weight subjects. Following VLCKD, a significant reduction in BMI and fasting blood glucose and lipids were observed as well as improved insulin sensitivity. The degree of liver steatosis only decreased in the subjects with severe steatosis. LAL activity significantly increased following VLCKD, however the level of activity was not significantly different either before or after dietary intervention compared to CG. The authors conclude that VLCKD induces LAL activity in morbidly obese subjects.
The attention to lysosomes and LAL activity associated with obesity, inflammation, and NAFLD is significantly on the rise. Therefore, the investigation of the level of activity of LAL following dietary intervention is important. However, the current state of the study presented here is rather preliminary. There is not enough data to draw any significant conclusions . . . yet. The data presented are strong for preliminary data sets, however, sufficient power and proper controls are needed. The two largest weakness are:
1) The authors conclude that findings are specific to VLCKD. This could very well be true however the experimental design is not convincing as the findings could be due to the calorie restriction (800 kcal/day) alone. Therefore, an obese subject group, also suitable for bariatric surgery, should be studied following a dietary intervention of calorie restriction in which the diet consist of a more balanced macronutrient composition.
Response #1
We agree with your concerns. However we think that our results are quite original to deserve publication, despite some limitations. In the discussion we clarify that our conclusions are just hypotheses, which need a confirmation with larger studies. The lack of a control group of obese subjects is already included among limitations of the study (Lines 329-334).
2) As no significant difference was observed in LAL activity either before or after VLCKD compared to the CG, the impact of the dietary intervention on LAL activity is completely lost. Increased power of analysis could rescue this major flaw.
Response #2
Our main hypothesis is an effect of VLCKD on LAL activity that is independent from weight loss. This hypothesis is clearly expressed in the discussion (Lines 309-313): “Furthermore, we should take into account the possibility of an independent effect of VLCKD on LAL activity. Indeed, its peculiar composition, particularly poor in carbohydrates and relatively rich in proteins, could modulate the liver metabolism of lipids acting on different molecular pathways, including LAL”. Pre-clinical studies, reported in the text, support this hypothesis (Lines 322-328): “This provisional conclusion is supported by the observation of the transcriptional effects of some essential amino acids in a murine model of the ketogenic diet. Indeed, previous studies have demonstrated that a diet enriched with five essential amino acids (Leu, Ile, Val, Lys, and Thr) prevents liver steatosis in mice fed with a high fat content diet [34] and it is associated with an increased expression of intracellular markers of mitochondrial function [35] and autophagy [36]”. This hypothesis could explain both the lack of statistical correlation between the weight loss and the increase of LAL activity, and the lack of a significant difference in LAL activity between patients and controls. The hypothesized U-shaped relationship between body weight and LAL activity is also consistent with this hypothesis; indeed, as it can be seen in Figure 2, patients and controls tend to lie on the two symmetrical branches of a parable. So, no significant difference can be detected between the two groups using a direct head-to-head confrontation.
As you correctly reported above, a larger number of enrolled patients and the presence of a second control group, composed of obese patients not undergoing VLCKD, would have stronger supported this hypothesis. We are already working to provide these data in future publications.
Comment #3
Of note, liver biopsies, collected at time of bariatric surgery, for histological and biochemical analysis could significant improve the impact of the study.
Response #3
We thank you for your valuable contribution. We will try to provide these results in future works, though performing liver biopsies without a strong clinical indication, in otherwise healthy patients, could raise some ethical concerns.
Comment #4
Furthermore, another major point of consideration is the use of the term ketogenic diet. The authors description of the diet is accurate as a very low carbohydrate diet will in fact generate ketones and a state of ketosis, as will calorie restriction, at least transiently. However, the ketogenic diet, in a general sense, is largely understood to be a low carbohydrate, high fat diet in which 20%, or less, of the calories are obtained from carbohydrates and 60%, or more, of the calories are obtained from fat. Therefore, it is highly recommended that authors reconsider using the term ketogenic. Although the ketogenic diet is trending in popularity, so is a low carbohydrate diet and is therefore of significant importance. The state of ketosis is however key to the current study as it suggest catabolic vs anabolic state of lipid metabolism.
Response #4
As we state in the Methods (Lines 129-133) “Self-testing of urinary ketones was administrated once daily. Urine ketone concentrations were measured using over-the-counter reagent strips (Accu-Chek Ketur Test, Roche Diagnostics GmbH, Mannheim, Germany), which determine the presence of ketones upon reaction with nitroprussiate salt. Urine ketones were assessed using a semi-quantitative scale. The compliance of patients to VLCKD was verified by the presence of ketones in the urine”. Since all patients had a positive test for urinary ketones up to 72 hours after the beginning of the diet, we are sure that all patients had a state of ketosis for the whole duration of the diet. This point was not clearly declared in the former version, so we added a sentence at Line 209: “All patients had a positive test for urinary ketones up to 72 hours after the beginning of the diet”
Comment #5
It is highly recommended that the authors seek out editing and writing services prior to submission of manuscript review.
The current version has been corrected by the language assistance service provided y the author. You can find the tracked changes in the text.
Comment #6
The current layout does not have an organized flow and is therefore challenging to follow especially for a reader outside this area of research.
Response #6
We used a layout template provided by the editor.
Comment #7
The introduction is insufficient. More detailed information should be provided for points of interest as opposed to list of reports of association. It currently reads as though there was no specific aim. Emphasis should be placed on the importance of LAL activity. Why should we care? What are clinical consequences of LAL activity expression? Is it merely a correlation or is it a potential pharmacological target?
Response #7
According to your suggestion, we tried to explain better our hypothesis and the aim of the study (Lines 68-75): “Since to our knowledge there are no data related to the modulation of LAL activity through the weight loss induced by VLCKD, we hypothesize that the effects of VLCKD on liver steatosis could be mediated by an induction of this enzyme. Therefore, the aim of our study was to evaluate the effect of a 25-day-long VLCKD on liver steatosis and LAL activity in a group of morbidly obese patients suitable for bariatric surgery. We also evaluated if a relationship exists between the effects of VLCKD on LAL activity and other anthropometric, cardiovascular, and metabolic modifications induced by VLCKD in morbidly obese subjects”
The potential clinical relevance of LAL activity in morbidly obese patients is explained in the discussion (Lines 309-318): “Furthermore, we should take into account the possibility of an independent effect of VLCKD on LAL activity. Indeed, its peculiar composition, particularly poor in carbohydrates and relatively rich in proteins, could modulate the liver metabolism of lipids acting on different molecular pathways, including LAL. This effect could explain the efficacy of VLCKD in reducing liver steatosis [31, 32]. Indeed, previous studies reported an association between reduced LAL activity and NAFLD [33], and consistently, we observed a significant improvement in the degree of liver steatosis after VLCKD, although no significant correlation between the improvement of steatosis degree and variation of LAL activity was observed. Moreover, this lack of significance could be explained by the limitations of the ultrasonographic measurement of liver steatosis, as discussed above”. Since this is one of the hypotheses that we intended to demonstrate, we couldn’t discuss this topic in the introduction. Furthermore, since we propose VLCKD as an effective tool for reducing the liver steatosis degree in morbidly obese patients, discussing the potential pharmacological treatments addressing LAL would be, in our opinion, beyond the aim of our study.
Comment #8
Table 1, Daily macronutrient composition would be better represented in an actual table format, listing macronutrient (carb, protein, and fat) content (mass and percentages) per meal and then totaled at bottom. This would make is clearer to the reader of the carefully executed experimental design.
Response #8
We agree with your comment. Diet composition was indeed poorly described; in particular caloric intake for each macronutrient was missing. Therefore, we implemented the description as follows (Lines 98-109): “Subjects underwent a 25-day planned VLCKD with caloric restriction (<800 kcal/day). Carbohydrate intake was <50 g/day (corresponding to <200 kcal/day) with a protein intake of 1.4 g/kg of ideal weight (calculated with the Lorentz formula [19]):
Ideal weight (male) = (height in cm – 100) - (height in cm – 150)/4;
Ideal weight (female) = (height in cm – 100) - (height in cm – 150)/2.
Assuming an average ideal weight of 65 kg for both women and men, the protein intake was >90 g/day (corresponding to >350 kcal/day). The remaining caloric intake was composed of fat (<250 kcal/day, corresponding to <30 g/day). The protein intake was achieved in part with a dietary supplement composed of milk whey protein (Nepicomplex, 39 g/day corresponding to 156 kcal/day) diluted in water or skimmed milk. The remaining part was achieved with fresh aliments in appropriate quantities, depending on ideal body weight. Details of the diet are shown in Table 1).
We preferred adding a description, rather than a table, because in our opinion a table would have been redundant and possibly confusing. We hope you will approve our decision.
Comment #9
Line 112: How were these parameters assessed? It is simply stated that they were measured. Please provide reference at minimum.
Response #9
We agree with your comment. We modified the sentence at Lines 136-140 in order to provide the information you require: “At baseline and after 25 days of VLCKD, the following parameters were evaluated: blood count (flow-cytometry), glutamic oxaloacetic transaminase (GOT), glutamic pyruvic transaminase (GPT), gamma glutamyl transpeptidase (γGT) (chemiluminescence), total and high-density lipoprotein (HDL)-cholesterol, triglycerides (mass spectrometry), fasting glucose, and insulin (immune-enzymatic assay)”
Comment #10
Also, would be better described if details (samples were collected following 12-h fast, etc) were mentioned here rather than previously in this section.
Response #10
Details of samples collection are referred not only to serum biochemical markers, but also to the DBS. Therefore, we preferred to put this sentence inside the general description of the methods.
Comment #11
Also, were parameters assessed when patients were still on diet or afterwards? Line 82 states “up to 7 days after end of diet.” This is important. If measurements were taken days after end of diet, why? Seems like this would confound results and any differences could be missed.
Comment #11
Measurements were taken before the beginning of the diet and after its end. Basically, measurements were collected one or two days after the diet, but this was not possible in every case because of patients’ own possibility to come to the office and perform the measurements. Therefore, we described the time lapse within all the measurements were surely completed.
We preferred to take the measurements after the end of the diet, rather than during the diet itself, because of the short duration of the diet. A shorter duration of observation would have prevented us from detecting some significant variations. We hope that you will comprehend the reason for our decision.
Comment #12
Line 248 - The authors state that a positive correlation was observed between LAL and markers of adiposity (body weight, BMI, waist circumference, and fat mass). Figure 2 only represents correlations in body weight and BMI yet the authors place emphasis on adiposity. The fat mass data from this study is available. If the authors wish to place emphasis on adiposity, a convincing correlation should be presented for fat mass as well.
Response #12
Figure 2 does not represent the correlations between LAL activity and the markers of adiposity, but the quadratic regression model between LAL activity and BMI/body weight. The correlations are only described in the text. Specifically the correlations are described at Lines 242-246 “After performing a univariate correlation analysis, a significant positive correlation was observed between LAL activity and body weight (r = .499, p = 0.025), BMI (r = .490, p = 0.028), waist circumference (r = .487, p = 0.029), and fat mass (r = .527, p = 0.020). A negatively significant correlation between LAL activity and body weight was also observed in the control group (r= -.563, p = 0.010)”.
Comment #13
5) Limitations to lack of significant differences in other parameters:
a. Low sample number
b. Short duration of diet
c. The fact that assessments were made up to 1 week after termination of the diet. Investigators may decrease the degree of variability if parameters are assessed while all patients are still on the diet. This would also simplify interpretation.
d. As the authors suggested, ultrasonographic measurement of liver steatosis.
e. Evidence that subjects remained in a daily state of ketosis. A low carbohydrate diet requires strict adherence to sustain ketosis.
Response #13
We partially agree with your comment. Most of the limitations you highlighted are indeed already declared in the text, i.e., low sample number and limitations of ultrasonographic measurement of liver steatosis.
Duration of ketogenic diet for the treatment of obesity lasts usually<30 days because of safety issues. Indeed, longer treatments are associated with an increased incidence of adverse effects, particularly kidney function impairment. See, for instance, Kosinski et al. Nutrients. 2017.
As stated in response to the comment #11, we preferred to take the measurements after the end of the diet, rather than during the diet itself, because of the short duration of the diet. A shorter duration of observation would have prevented us from detecting some significant variations.
As stated in response to Comment #4 we added a sentence at Line 204: “All patients had a positive test for urinary ketones up to 72 hours after the beginning of the diet”.
Round 2
Reviewer 2 Report
It is well known that the very low carbohydrate ketonic diet can have adverse health effects and in general such diet should not be advised, however, it can be useful tool to decrease body weight when performed under professional supervision for a relatively short time. Therefore, more reflection is expected and the Authors are still expected to present such reflection not simply present VLCKD diet as” an emerging technique to induce a significant, well-tolerated, and rapid loss of body weight in morbidly obese patients”. This problem should be presented in the Introduction and further under Discussion.
The content of vitamins and minerals in the supplements should be presented
In the Method section ,no references are provided. It is indicated that total and high-density lipoprotein (HDL)-cholesterol, triglycerides were assessed by mass spectrometry, as it is not a method commonly used in clinical practice more information and/or references are needed.
Reviewer 3 Report
My first comment has not been addressed sufficiently. LAL activity increased in the subgroup of patients after VLCD. However, an obese control group was not included. It cannot be excluded that he changes in LAL activity are due to variation in time, the VLCKD or the weight loss.
An experiment with an intervention, but without a parallel control group is difficult to interpret. This is an important flaw in the design of the study.
It is also not clear why no significant difference was observed between obese patients and healthy controls in LAL activity. Therefore, the relevance and reliability of the LAL measurements is not clear.
The authors only state that there were no differences between obese patients and healthy controls in LAL activity. However, it is unclear to me how this should be interpreted. If body weight is important in LAL acytivity, why is there no difference between obese patients and healthy controls?
Reviewer 4 Report
N/A